# Ultrasonographic Evaluation of Entheseal Fibrocartilage in Patients with Psoriatic Arthritis, Athletes and Healthy Controls: A Comparison Study

**DOI:** 10.3390/diagnostics13081446

**Published:** 2023-04-17

**Authors:** Fabio Massimo Perrotta, Mario Ronga, Silvia Scriffignano, Ennio Lubrano

**Affiliations:** 1Dipartimento di Medicina e Scienze della Salute “Vincenzo Tiberio”, Università degli Studi del Molise, 86100 Campobasso, Italy; 2Orthopedics and Trauma Operative Unit, Department of Health Sciences, University of Piemonte Orientale, 13100 Novara, Italy; 3Department of Development and Regeneration, Skeletal Biology and Engineering Research Center, KU Leuven University, 3000 Leuven, Belgium

**Keywords:** ultrasonography, enthesis, fibrocartilage, psoriatic arthritis

## Abstract

The aims of this study were as follows: (1) To evaluate the entheseal fibrocartilage (EF) during Achilles tendon insertion in patients with Psoriatic Arthritis (PsA) by using power Doppler ultrasound (PDUS), (2) to assess the intra and inter-reader reliability of the evaluation of EF thickness, (3) to compare the EF thickness of PsA patients, athletes and healthy controls (HCs), and (4) to evaluate the correlations between EF abnormalities, disease activity and functional indices in PsA. Methods: Consecutive PsA patients attending our unit were asked to participate. HCs and agonist athletes were enrolled as a control group. A bilateral PDUS evaluation of Achilles tendons was performed in order to evaluate the EF in all patients and controls. Results: In total, 30 PsA patients, 40 athletes and 20 HCs were enrolled. The median (IQR) EF thickness among the PsA patients, athletes and HCs was 0.035 cm (0.028–0.04) cm, 0.036 (0.025–0.043) cm and 0.030 (0.020–0.038) cm, respectively (*p* = 0.05 between PsA patients and HCs). The intra-reader reliability was excellent [ICC (95% CI) of 0.91 (0.88–0.95)] and the inter-reader reliability was good (0.80 (0.71–0.86). The assessment of EF was feasible, with a mean time of 2 min. No correlations were found with disease activity indices in PsA patients. Conclusion: The assessment of EF is a feasible and reproducible test and may be explored as a potential imaging biomarker.

## 1. Introduction

Psoriatic Arthritis (PsA) is a chronic inflammatory disease that is characterized by the presence of arthritis, enthesitis, dactylitis and axial involvement in patients with a personal or family history of psoriasis, and is a disease in which different clinical manifestations run together [1,2]. Of the different clinical features of this intriguing disease, enthesitis can be considered the hallmark of PsA and of the whole spondylarthritis (SpA) group in general. Peripheral enthesitis is usually revealed by clinical findings, such as localized pain, tenderness and swelling. However, imaging techniques, and in particular, the use of power Doppler ultrasonography (PDUS), can be used to improve clinical examination [3,4,5]. PDUS was found to perform better than clinical examination for detecting entheseal abnormalities, but there is a considerable discrepancy between the clinical and PDUS findings, with less than 50% of clinically tender entheses being related to inflammatory enthesitis when assessed using ultrasound [6]. On the other hand, entheseal abnormalities could be found in asymptomatic PsA, in patients with early PsA and in psoriasis alone [7,8,9,10].

Recently, OMERACT published a scoring system for enthesitis that assesses the presence/absence of hypoechogenicity, thickening, calcifications/enthesophytes, and erosions, as well as color Doppler activity semi-quantitatively expressed from 0 to 3. This tool allows the assessment of enthesitis in the whole group of SpA to be better evaluated and standardized [11].

On the other hand, from a pathogenetic point of view, improvements in PDUS imaging techniques may further enhance our capacity to evaluate the inflamed target tissue in SpA and PsA [12,13,14]. In particular, various studies have shown how the concept of the “synovial–entheseal organ” has a central role in the pathogenesis of SpA, in which the pathological process occurs mainly at fibrocartilaginous enthesis; here, the fibrocartilaginous tissue is the main site of inflammatory cell infiltration [15,16].

Entheseal fibrocartilage (EF) can be assessed using different imaging tools, including magnetic resonance imaging (MRI) and PDUS. However, the disadvantages of MRI include patient discomfort, the lack of availability, and time and cost; meanwhile, the availability of high-resolution PDUS can ensure a good visualization of this important tissue, which can be visualized as an anechoic layer at the enthesis, immediately adjacent to the bone [17,18].

The validity of using PDUS to visualize EF in terms of face and content was demonstrated in one study [17], but the structural changes in EF in PsA have not been extensively studied and no information is available regarding possible correlation of EF with clinical disease activity, functional indices or with the presence of other active or structural changes at entheseal sites.

Furthermore, few studies have been published on the modifications made to fibrocartilage during growth, aging processes or in biomechanical stress conditions [19,20]; therefore, there is an unmet need regarding this important aspect.

In this light, the aims of this study were as follows: (1) to evaluate the EF during Achilles tendon insertion by using PDUS in PsA patients, (2) to assess the intra and inter-reader reliability of the evaluation of EF thickness, (3) to compare the EF thickness of PsA patients, athletes and healthy controls (HCs), and (4) to evaluate the correlation of EF thickness with disease activity and functional indices in PsA.

## 2. Materials and Methods

A cross-sectional study that included consecutive PsA patients who attended our Unit (Rheumatology Unit, University of Molise), fulfilled the inclusion criteria and were asked to participate was conducted. Furthermore, soccer and basketball players attending the Sport Medicine clinic of our department for routine controls that fulfilled the inclusion criteria, and a group of HCs, were asked to participate and were enrolled as control groups. All patients and controls were enrolled from March 2022 to October 2022.

The inclusion criteria for the PsA patients were as follows:(1)Age < 50 years-old.(2)Satisfying CASPAR criteria for PsA [21].

The inclusion criteria for the athletes were as follows:(3)At least 5 years of competitive activity and at least 10 h of weekly training.(4)Absence of any musculoskeletal complaints in recent years.

The exclusion criteria included participants of the age < 18 years old.

Demographic and clinical data, including age, sex, height, weight, body mass index (BMI), smoking habits and hours of weekly training (for the athletes group), were collected for each patient and the controls. Finally, it was ensured that PsA patients had not been athletes in the past.

### 2.1. Clinical Assessment of PsA Patients

A detailed medical history and physical examination data were collected for all patients. Demographics and disease characteristics, including disease duration, level of education and pattern of articular manifestations, were evaluated. The laboratory parameters included the C-reactive protein test (performed within two weeks of enrollment) as a part of the routine monitoring of our patients. The clinical assessment was performed by the same expert rheumatologist (EL) and encompassed an evaluation of 68 tender and 66 swollen joints, experiencing enthesitis and dactylitis. The pattern of articular involvement was also collected, as well as comorbidities and related diseases. Enthesitis was assessed by using the Leeds Enthesitis Index (LEI) [22], and dactylitis was determined as present/absent. Skin assessment was performed using the Body Surface Area (BSA). The Patient Global Assessment (PtGA), pain assessment on the Visual Analogic Scale (VAS) and the physician’s global evaluation of disease activity on a VAS scale were also recorded [23]. The disease status and disease activity were assessed using the Minimal Disease Activity criteria (MDA) [24] and the Disease Activity score for Psoriatic Arthritis (DAPSA) [25,26]. The patient acceptable symptom state (PASS) was also collected [27]. The Psoriatic Arthritis Impact of Disease (PsAID) [28] was evaluated as measures of function and the patients’ quality of life. Information on the patients’ previous and current use of conventional synthetic, biologic/target synthetic disease-modifying anti-rheumatic drugs (DMARDs) was collected.

### 2.2. Ultrasound Assessment

Patients, athletes and HCs underwent a PDUS evaluation during bilateral Achilles tendon insertions. PDUS assessments, using the grey scale and power Doppler modes, were performed in a darkened room by two rheumatologists (FMP and SS) using the same equipment, who were blinded to the study groups and to each other’s findings. Patients were asked to take a prone position with their feet hanging over the examination table in a neutral position for the assessment of the Achilles tendon insertion. Bilateral ultrasound examination was carried out using a multiplanar scanning technique. Particular attention was paid to minimizing the transducer pressure on the anatomical structures by using a large amount of gel. Care was also taken to alter the angle of the probe in order to prevent anisotropy. High-resolution US assessment was performed using a Samsung HM70A plus device with linear 6–18 MHz probes. For PD examinations, the pulse repetition frequency was set to 800 Hz. Images were electronically stored and blinded in order to determine the time point of acquisition and a given subject. To assess the intra-reader reliability, images were re-assessed after two weeks.

Findings consistent with entheseal pathology were graded both on a binary scale in the grey scale mode and semi-quantitatively from 0 to 3 in the power Doppler mode, as previously described [11]. Furthermore, the presence of tendon calcifications, enthesophytes, bone erosions, infrapatellar bursitis, hypoechogenicity and increased enthesis thickness was evaluated according to OMERACT and as described elsewhere [11]. Figure 1 describes some of the PDUS findings in our patients.

### 2.3. Ultrasound Assessment of EF

The thickness of the anechoic layer was measured at its thickest point in the longitudinal scan (Figure 1a), as previously described [17].

EF abnormalities, such as focal disappearing and erosions of the cortical bone, were also evaluated.

### 2.4. Ethical Approval

The study was approved by the institutional review board of the University of Molise (protocol n. 0001-017-2021) and performed according to the Helsinki declaration. Written informed consent was obtained for all patients and controls regarding their participation in the study and the use of clinical data.

### 2.5. Statistical Analysis

Statistical analysis was performed using IBM SPSS Statistics for Macintosh, Version 28.0.

Numerical variables were summarized by using the median and Inter-Quartile Range (IQR), or the minimum and maximum value (min/max).

Intra and inter-reader reliability were assessed by using the intraclass correlation coefficient-ICC (95% confidence interval-CI).

For continuous variables, the significance of the differences was determined using the Student’s *t* test for unpaired data for variables normally distributed, and the Mann–Whitney test was used for unpaired non-normally distributed variables. We used the Shapiro–Wilk test to evaluate the distribution of the variables.

Correlation between the EF thickness (mean between the two sites) and clinical features was evaluated using Spearman’s Rho. Kruskal–Wallis analysis was carried out to evaluate the differences in EF among the three groups. Dunn’s post hoc tests were carried out on each pair of groups. Categorical variables were compared using the χ-square test or Fisher’s exact test. A *p* ≤ 0.05 value was deemed statistically significant.

## 3. Results

We enrolled 30 PsA patients, 40 athletes and 20 HCs. A total of 178 enthesis sites were evaluated (one athlete had missing data).

Of the 30 PsA patients, 4 (13.3%) reported previous Achillodynia. The others did not report entheseal symptoms in their clinical histories.

Table 1 summarizes the clinical characteristics of PsA patients, athletes and HCs.

During PDUS examination, no statistically significant differences were found between the two sides (right and left within each group).

The intra-reader reliability was excellent [ICC (95%CI) of 0.91 (0.88–0.95)] and the inter-reader reliability was good [ICC (95% CI) of 0.80 (0.71–0.86)]. Table 2 shows the intra and inter-reader reliability for both sides (single measures and average level).

Figure 2 shows the EF thickness differences (Kruskal–Wallis) among the PsA patients, HCs and athletes. The median (IQR) EF thickness among the PsA patients, athletes and HCs was 0.035 cm (0.028–0.04) cm, 0.036 (0.025–0.043) cm and 0.030 (0.020–0.038) cm, respectively (*p* = 0.05 between PsA and HCs; *p* = 0.008 between athletes and HCs). PsA patients and athletes had a statistically significant higher EF thickness with respect to HCs. Moreover, no differences were found between the sexes in terms of the EF thickness.

Interestingly, no statistically significant correlations were found between the EF thickness and BMI, age, disease duration, LEI, DAPSA, and PsAID in PsA, and no correlations were found between clinical characteristics and the hours of training per week in athletes (Table 3).

Of note, during the PDUS examination of the enthesis, erosions and power doppler positivity within 2 mm from the cortical bone were present in four of the PsA patients. Erosions were not found in athletes and HCs. In the four PsA patients with cortical bone erosions, EFIwas not visible within the erosion. One HC showed power doppler positivity at the entheseal site. Enthesophytes were generally more present in athletes. Figure 3 shows the PDUS findings at entheseal sites by groups.

## 4. Discussion

Enthesitis is the hallmark of SpA, including PsA. Inflammation at entheseal sites also affects the bone and other structures, such as bursae, and supports the concept of a synovial–entheseal complex that has been proposed by McGonagle et al. in recent years [29,30]. The essential importance of the enthesis in understanding the pathophysiology of inflammation and structural damage in PsA has reemerged in front of the mechanical overload concept, which is linked to the pathogenesis of enthesopathy [31,32]. Moreover, the link between enthesitis and osteitis in SpA has been clarified in recent studies, which have demonstrated the close functional integration of the enthesis with the neighboring bone, with the EF playing a central role [29]. In this context, inflammatory changes may be translated into EF abnormalities, which could be directly assessed in vivo by using imaging modalities, in the context of other well-characterized inflammatory changes in the enthesis, such as hypoechogenicity and entheseal thickening. However, no formal studies on the EF have been performed, except for the study published by Aydin et al. [17], who found no differences in the EF in a small group of SpA patients compared to HCs. In this interesting preliminary study, the authors aimed to demonstrate the face and content validity of using ultrasound for EF visualization, both by bovine histological evaluation and EF imaging in SpA. The authors found no differences in the EF thickness, but an interruption of EF, in the case of cortical bone erosion, was observed. Furthermore, male and female patients showed different entheseal thicknesses and some data showed that males have a greater EF thickness than females [33].

Our results partially agree with those previously published. In particular, some differences and similarities have been found: (1) the median EF thickness values reported in our study are slightly lower than those reported by Aydin et al. This could be related to the different US equipment used; (2) the measure of EF was feasible and reliable and could be easily performed in clinical practice; and (3) our study confirms the EF interruption when erosive changes are present but, interestingly, also demonstrates a slight but statistically significant increase in the EF thickness in PsA patients and athletes compared to HCs.

This result is of particular interest, since, in patients without clinically evident enthesitis, changes in EF could be observed, resembling those seen in agonist athletes; however, the clinical significance of this result must be further evaluated.

Furthermore, no patients had clinically active enthesitis and most of the patients were in good control of the disease. This, in turn, may be the reason for the lack of correlation between EF thickness and the composite measure of disease activity. It could be speculated that EF thickness represents a marker of damage in athletes due to repeated trauma while that of long-term damage in PsA patients.

We also confirm the reproducibility of the EF assessment using PDUS examination with good values of ICC found regarding the intra and inter-reader reliability.

Moreover, the assessment of EF was feasible and quick, and required no more than 2 min.

In this light, PDUS of EF could be integrated into the evaluation of the enthesis organ and may provide further information regarding both PsA and SpA, in which clinical and PDUS findings do not always overlap [34,35].

This study has some limitations: the cross-sectional design does not permit the assessment of sensitivity to change, and thus further studies are needed. Moreover, we did not enroll patients with degenerative diseases such as osteoarthritis, which may produce abnormalities in the EF. Finally, the median age and sex were not balanced between the three groups; however, no correlation was found between EF thickness and age.

## 5. Conclusions

In conclusion, this study has demonstrated the reliability of the evaluation of EF thickness and significant differences in respect to HCs. However, we fail to demonstrate a correlation with the disease activity indices, probably due to the factors mentioned above.

Our pilot study potentially suggests the use of EF assessment in the evaluation of PDUS abnormalities in PsA patients and may pave the way for further large studies on this particular aspect.

## Figures and Tables

**Figure 1 diagnostics-13-01446-f001:**
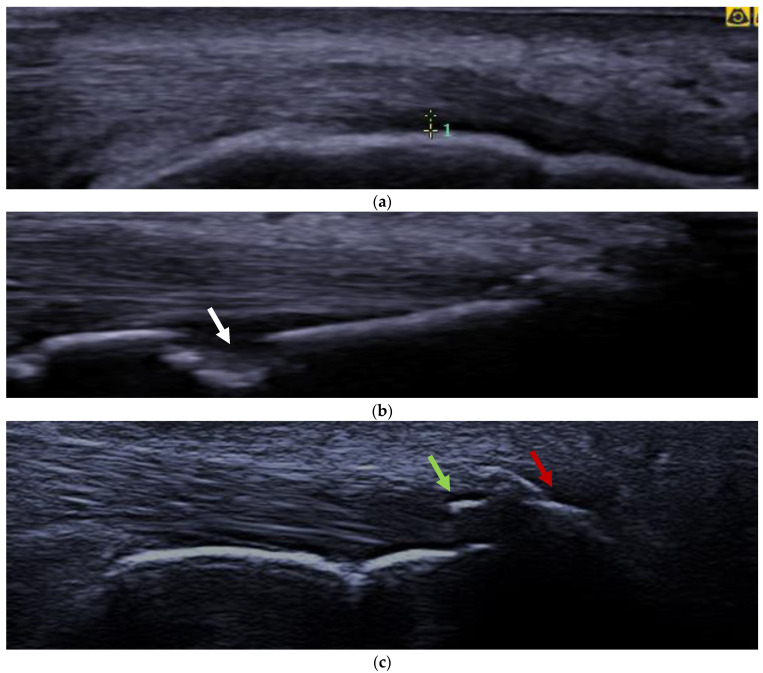
(**a**) Measure of EF thickness at Achilles tendon insertion. (**b**) Erosion (white arrow) at Achilles tendon insertion in a PsA patient. (**c**) Enthesophyte (red arrow) and calcification (green arrow) at Achilles tendon in athletes.

**Figure 2 diagnostics-13-01446-f002:**
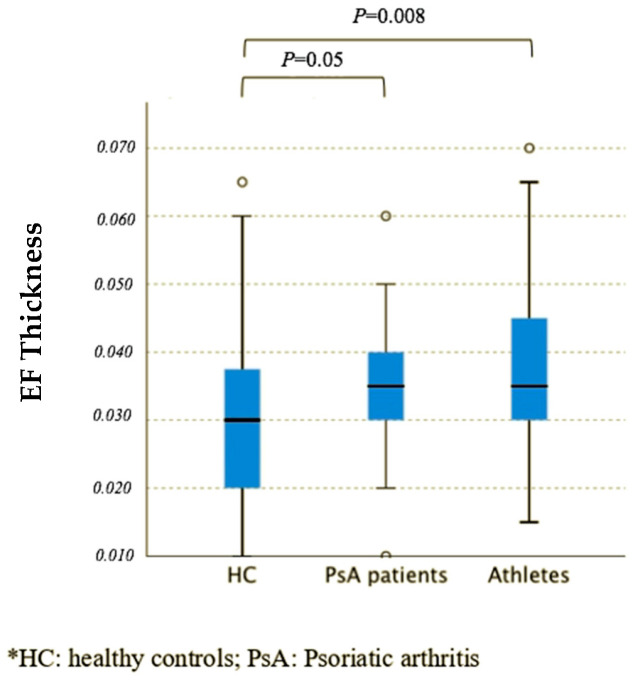
Comparison of the median EF thickness (pooled data of right and left side) between the three groups (independent samples Kruskal–Wallis test).

**Figure 3 diagnostics-13-01446-f003:**
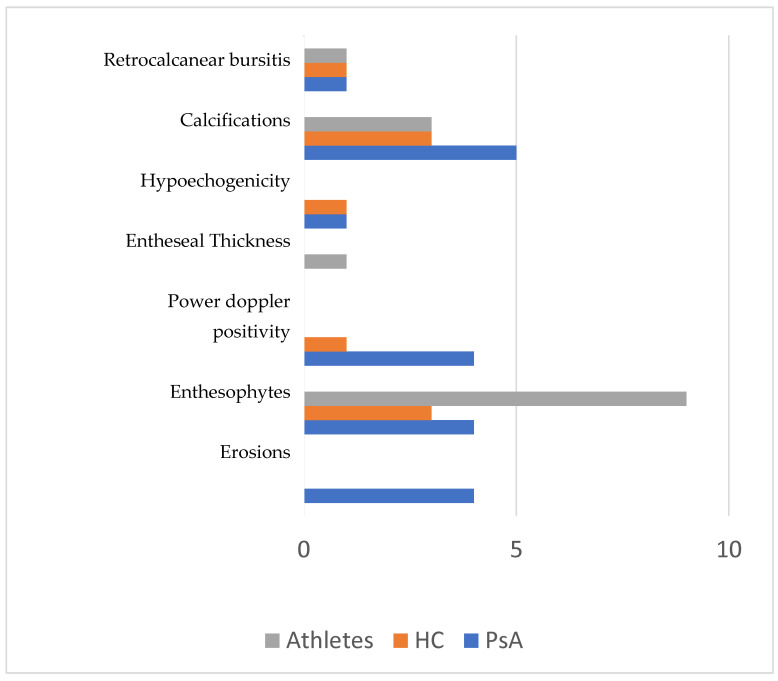
Ultrasound findings by group.

**Table 1 diagnostics-13-01446-t001:** Clinical and imaging characteristics of the enrolled subjects.

	PsA Patients n. 30	Athletes n. 40	Healthy Control n. 20
Sex (F), n. (%)	7 (23.3)	12 (30)	10 (50)
Age, median (range)	48 (22–50)	24 (20–41)	25.3 (21–40)
Weight, Kg, median (IQR)	80 (72–88)	70 (65–74)	63.5 (58–70)
Height, m, median (IQR)	1.74 (1.72–1.76)	1.75 (1.69–1.80)	1.67 (1.63–1.75)
BMI, median (IQR)	26.12 (23.70–29.22)	23.10 (22.18–24.55)	21.71 (19.59–23.26)
Smoke (yes), n. (%)	4 (14.3)	7 (18.4)	7 (31.8)
Enthesopathy (yes), n. (%)	6/26 (23.1)	13 (34.2)	6 (27.3)
Entheseal calcifications (yes), n. (%)	8/28 (28.6)	11 (28.9)	7 (31.8)
EF thickness (right), cm, median (IQR)	0.0350 (0.0300–0.0400)	0.0350 (0.0300–0.0413)	0.0300 (0.0200–0.0400)
EF thickness (left), cm, median (IQR)	0.0350 (0.0263–0.0400)	0.0375 (0.0200–0.0450)	0.0300 (0.0200–0.0363)
DAPSA, median (IQR)	5.75 (4.00–14.26)	-	-
LEI, median (IQR)	0 (0–0)	-	-
Disease duration (months), median (IQR)	48 (12.5–79)	-	-
BSA, median, (IQR)	0 (0–1)	-	-
Hours of training (per week), median (IQR)	-	12 (8–15)	

BMI: body mass index; EF: entheseal fibrocartilage; IQR: interquartile range; DAPSA: disease activity index for psoriatic arthritis; LEI: Leeds enthesitis index; BSA: body surface area.

**Table 2 diagnostics-13-01446-t002:** Intra and inter-reader reliability; intraclass correlation coefficient (95% confidence interval).

EC Thickness (Right)	Single Measures	*p* Value	Average Measures	*p* Value
**Intra-reader**	0.910 (0.880–0.950)	<0.001	0.950 (0.890–0.970)	<0.001
**Inter-reader**	0.802 (0.713–0.866)	<0.001	0.890 (0.833–0.928)	<0.001
**EC thickness (left)**				
**Intra-reader**	0.910 (0.880–0.950)	<0.001	0.920 (0.890–0.960)	<0.001
**Inter-reader**	0.787 (0.693–0.855)	<0.001	0.881 (0.819–0.922)	<0.001

EF: entheseal fibrocartilage.

**Table 3 diagnostics-13-01446-t003:** Correlation between EF thickness and clinical characteristics in our group of patients.

EF Thickness (Right)		
	rho	*p* value
**All (PsA + HCs + Athletes)**		
Age	−0.144	0.177
Weight	0.193	0.072
BMI	0.117	0.277
**PsA patients**		
DAPSA	0.194	0.364
Disease duration (months)	−0.171	0.375
PsAID	0.200	0.240
LEI	0.020	0.780
**Athletes**		
Hours of training per week	0.210	0.120
**EF thickness (left)**		
	rho	*p* value
**All (PsA + HCs + Athletes)**		
Age	−0.091	0.398
Weight	0.156	0.148
BMI	0.149	0.165
**PsA patients**		
DAPSA	0.036	0.868
Disease duration (months)	−0.094	0.635
PsAID	0.210	0.300
LEI	0.140	0.460
**Athletes**		
Hours of training per week	0.150	0.172

EF: entheseal fibrocartilage; PsA: psoriatic arthritis; HCs: healthy controls; BMI: body mass index; DAPSA: disease activity score for psoriatic arthritis; PsAID: psoriatic arthritis impact of disease; LEI: Leeds enthesitis index.

## Data Availability

Datasets generated for the study are available upon reasonable request.

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
