# Peer review of "Ultrasonographic Evaluation of Entheseal Fibrocartilage in Patients with Psoriatic Arthritis, Athletes and Healthy Controls: A Comparison Study"

_diagnostics, 2023, doi:10.3390/diagnostics13081446_

Round 1
Reviewer 1 Report
I read with interest the paper from Perrotta et al, who explored the US assessment of EF in a mixed cohort of PsA patients, athletes and HC.
Despite the interesting evaluation, I see again more a confirmation of the face validity of the technique, than a real potential role for this assessment in clinical practice.
These are my comments:
- if one athlete has missing data, then why not excluding?
- as a limitation, HC were not matched also for gender, not only for sex. In addition, we must assume that patients were not athlete in the past, given their older age.
- correct typo in table 2 "itra"
- I would consider analyzing the role of possible predictors of EF thickness with a linear regression (combining left and right measurements and adjusting for repeated measures). In this case, you could use age, sex and weight to adjust the three groups (considered as a single variable) and use PsA patients as a control. This would actually allow to distinguish a possible role for this evaluation in PsA versus athletes.
- the PsA population is quite well controlled in terms of disease activity, 25% in remission, almost 50% in low disease activity. Even more, the number of patients with LEI score >0 is definitely small (potentially 0), in particular it might not be clear if PsA patients had entheseal involvement and specifically of the Achilles tendons. This might be a big limitation to the possibility of using this as a marker of disease activity and possibly explaining the lack of correlation.
- in addition, given the similarity between athletes and PsA patients, would you speculate that EF thickness could probably represent a marker of damage, instead of disease activity? this could be the possible result of repeated trauma in athletes while of long term damage in PsA.
Author Response
Response to reviewer’s comments
Reviewer n.1.
I read with interest the paper from Perrotta et al, who explored the US assessment of EF in a mixed cohort of PsA patients, athletes and HC.
Despite the interesting evaluation, I see again more a confirmation of the face validity of the technique, than a real potential role for this assessment in clinical practice.
These are my comments:
- if one athlete has missing data, then why not excluding?
- We want to thank the reviewer for the comment. We reported all data in table one and we excluded the patients from the analysis.
- as a limitation, HC were not matched also for gender, not only for sex. In addition, we must assume that patients were not athlete in the past, given their older age.
- We do agree with the reviewer. Of course, it could be a limitation of the study. We further discussed this point in the discussion section by reporting the that HC were not age and sex matched. Furthermore, all PsA patients were not athlete in the past. We added this information in the manuscript.
- correct typo in table 2 "itra"
- Thanks for the suggestion. We corrected the typo in table 2
- I would consider analyzing the role of possible predictors of EF thickness with a linear regression (combining left and right measurements and adjusting for repeated measures). In this case, you could use age, sex and weight to adjust the three groups (considered as a single variable) and use PsA patients as a control. This would actually allow to distinguish a possible role for this evaluation in PsA versus athletes.
- We partially agree with the reviewer, because linear regression would bring interesting information. However, we assessed the assumption to perform linear regression and, considering the results of the correlations, which showed no linear correlation, we did not perform the linear regression.
- the PsA population is quite well controlled in terms of disease activity, 25% in remission, almost 50% in low disease activity. Even more, the number of patients with LEI score >0 is definitely small (potentially 0), in particular it might not be clear if PsA patients had entheseal involvement and specifically of the Achilles tendons. This might be a big limitation to the possibility of using this as a marker of disease activity and possibly explaining the lack of correlation.
- We do agree with the reviewer, in fact, our group of patients were in a good control of the disease (all patients were in biologic treatment). In the results section, we reported that no patients had clinically active enthesitis and this, of course, may be the reason for the lack of correlation. We further discussed this point in the discussion section
- in addition, given the similarity between athletes and PsA patients, would you speculate that EF thickness could probably represent a marker of damage, instead of disease activity? this could be the possible result of repeated trauma in athletes while of long term damage in PsA.
- We do agree with the reviewer and we further discussed this intriguing point in the discussion section.
Reviewer 2 Report
1. Introduction. The authors write that „Psoariatic Artrhtis (PsA) is a chronic inflammatory disease characterized by the association of psoriasis and arthritis”. The statement is true for most cases, in which psoriasis and arthritis are associated, but there are cases of PsA without psoriasis (sine psoriasis). See for example DOI: 10.3899/jrheum.090218.
2. Introduction. The authors write that “no information are available in literature, to our knowledge, on the possible correlation with clinical disease activity or functional indices”. How did this knowledge come about? The authors should add objectivity to this statement and declare, if true, that they searched PubMed, Web of Science, Scopus et al. and found no such information.
3. Inclusion criteria. Are the patients and controls adults or did the authors include children or adolescents? A minimum inclusion age should appear in the text as an inclusion or exclusion criterion.
4. PsA evaluation. The authors included CRP measurements. What kind of CRP (maybe it was high sensitivity CRP)? What method? Were all the CRP measurements done by the same lab?
5. Ethics. The authors wrote that “written informed consent to use clinical data of all participants was obtained.” It is good that they agreed to give their clinical data, but the must have also agreed to participate in the study. The statement that they agreed to participate in the study must be present in the text.
6. Statistics. SPSS is not cited properly. Please see: https://www.ibm.com/support/pages/how-cite-ibm-spss-statistics-or-earlier-versions-spss.
7. Statistics. There are 2 problems regarding the tests. First, the authors rightly state that they used different statistical tests depending on the normality of distribution. They should declare how did they evaluate the normality of distribution. Second, they chose to report their variables as medians with IQR/min-max. By choosing medians, it means that all their continuous variables were non-normally distributed and, in consequence, that they should not have used parametric tests such as the t test. If some of their variables were actually normally distributed, these variables should be reported as means with standard deviations.
8. Results. The results section starts with the time interval of inclusion (“March 2022 to October 2022”). This is not a result; it is a study design characteristic and should be moved in the Methods section.
9. Results. The phrase “The assessment of EF was feasible and quick and requires no more than 2 minutes.” is present between the results. This should be deleted or moved to the Introduction or the Discussion section.
10. Results. The authors report in the text and in Figure 2 the EF thickness differences tested with a Kruskall-Wallis test and comparisons between pairs of the 3 subgroups as outlined by different p values. This means that the also did a post-hoc analysis of the Kruskall-Wallis test, but the Statistics section does not mention it. If this is the case, the Statistics section should include this information on the use of a post-hoc test and which one.
11. Conclusion. The conclusion is incomplete, since it simply says that EF can be assessed with PDUS in PsA. Why? The authors should briefly touch their nicely worded objectives of the study, namely objective 2 (did they find reliability of EF assessment?), objective 3 (is EF thickness different in PsA compared to A and HC?) and objective 4 (does EF thickness correlate with anything?).
12. The text has English shortcomings and should be revised by a specialist. However, I will address 2 wording problems that may alter the scientific content of the text:
a. Introduction. The authors write that the “tool allows to better evaluate and to uniform the assessment of enthesitis”. The verb “to standardize” instead of “to uniform” is a much better fit.
b. Results. The authors write between parathesis that “on athlete has missing data”. They probably intended to use a numeral, so it’s “one” = 1.
Author Response
Response to Reviewers' comments
Reviewer n. 2
Introduction. The authors write that „Psoariatic Artrhtis (PsA) is a chronic inflammatory disease characterized by the association of psoriasis and arthritis”. The statement is true for most cases, in which psoriasis and arthritis are associated, but there are cases of PsA without psoriasis (sine psoriasis). See for example DOI: 10.3899/jrheum.090218.
R: We would like to thank the reviewer for the comment. We do agree and we modified the text by adding the suggested paper
- Introduction. The authors write that “no information are available in literature, to our knowledge, on the possible correlation with clinical disease activity or functional indices”. How did this knowledge come about? The authors should add objectivity to this statement and declare, if true, that they searched PubMed, Web of Science, Scopus et al. and found no such information.
R: we do agree with the reviewer and we modified the text. We deleted the aforementioned sentence.
- Inclusion criteria. Are the patients and controls adults or did the authors include children or adolescents? A minimum inclusion age should appear in the text as an inclusion or exclusion criterion.
R: We do agree with the reviewer, as you can see in table 1, all patients and controls were adult. We addedd, in the methods section, the age < 18 years as an exclusion criterion.
- PsA evaluation. The authors included CRP measurements. What kind of CRP (maybe it was high sensitivity CRP)? What method? Were all the CRP measurements done by the same lab?
- We would like to thank the reviewer for the comment. This is a clinical study performed in a context of clinical practice in which CRP measurement is part of the routine evaluation of PsA patients and it is included in the composite disease activity index DAPSA. For the purpose of the study, CRP method of evaluation was not reported. Furthermore, CRP serum levels coming from different labs. We usually prescribe, to our patients, the determination of high sensitivity CRP.
We hope we have addressed this point.
- Ethics. The authors wrote that “written informed consent to use clinical data of all participants was obtained.” It is good that they agreed to give their clinical data, but the must have also agreed to participate in the study. The statement that they agreed to participate in the study must be present in the text.
- We do agree with the reviewer and we modified the text by reporting that the patients signed to agree to partecipate in the study
- Statistics. SPSS is not cited properly. Please see: https://www.ibm.com/support/pages/how-cite-ibm-spss-statistics-or-earlier-versions-spss.
R: Thanks for the comment. We modifiy the text accordingly.
- Statistics. There are 2 problems regarding the tests. First, the authors rightly state that they used different statistical tests depending on the normality of distribution. They should declare how did they evaluate the normality of distribution. Second, they chose to report their variables as medians with IQR/min-max. By choosing medians, it means that all their continuous variables were non-normally distributed and, in consequence, that they should not have used parametric tests such as the t test. If some of their variables were actually normally distributed, these variables should be reported as means with standard deviations.
R: We do agree with the reviewer, we modified the text in the statistical methods section. We confirm that all variable were non-normally distribuited and, for this reason we used the median/IQR range. We used the Shapiro-Wilk test to assess the normality. We added this sentence in the statistical methods section.
- Results. The results section starts with the time interval of inclusion (“March 2022 to October 2022”). This is not a result; it is a study design characteristic and should be moved in the Methods section.
- We do agree with the reviewer and we modified accordingly
- Results. The phrase “The assessment of EF was feasible and quick and requires no more than 2 minutes.” is present between the results. This should be deleted or moved to the Introduction or the Discussion section.
R: We do agree and we modified accordingly
- Results. The authors report in the text and in Figure 2 the EF thickness differences tested with a Kruskall-Wallis test and comparisons between pairs of the 3 subgroups as outlined by different p values. This means that the also did a post-hoc analysis of the Kruskall-Wallis test, but the Statistics section does not mention it. If this is the case, the Statistics section should include this information on the use of a post-hoc test and which one.
R: We do agree with the reviewer. We performed the Dunn post hoc-test. We reported this infotmation in the statistical methods section
- Conclusion. The conclusion is incomplete, since it simply says that EF can be assessed with PDUS in PsA. Why? The authors should briefly touch their nicely worded objectives of the study, namely objective 2 (did they find reliability of EF assessment?), objective 3 (is EF thickness different in PsA compared to A and HC?) and objective 4 (does EF thickness correlate with anything?).
R: We would like to thank the reviewer for the comment which help us to improve the quality of the manuscript. We do agree with the reviewer and we further discussed our results in the discussion section, as suggested.
- The text has English shortcomings and should be revised by a specialist. However, I will address 2 wording problems that may alter the scientific content of the text:
- Introduction. The authors write that the “tool allows to better evaluate and to uniform the assessment of enthesitis”. The verb “to standardize” instead of “to uniform” is a much better fit.
- Results. The authors write between parathesis that “on athlete has missing data”. They probably intended to use a numeral, so it’s “one” = 1.
R: We would like to thank the reviewer for the comment which help us to improve the quality of the manuscript.We modified the sentences as suggested and we improved the quality of the English language.
Round 2
Reviewer 1 Report
The authors replied all comments, no further suggestion from my side. Thank you